# Achieving an Excellent Strength and Ductility Balance in Additive Manufactured Ti-6Al-4V Alloy through Multi-Step High-to-Low-Temperature Heat Treatment

**DOI:** 10.3390/ma16216947

**Published:** 2023-10-29

**Authors:** Changshun Wang, Yan Lei, Chenglin Li

**Affiliations:** School of Power and Mechanical Engineering, Wuhan University, Wuhan 430072, China; 2020282080098@whu.edu.cn

**Keywords:** Ti-6Al-4V, selective laser melting, post-heat treatment, high strength and ductility, isotropy

## Abstract

Selective laser melting (SLM) can effectively replace traditional processing methods to prepare parts with arbitrary complex shapes through layer-by-layer accumulation. However, SLM Ti-6Al-4V alloy typically exhibits low ductility and significant mechanical properties anisotropy due to the presence of acicular α′ martensite and columnar prior β grains. Post-heat treatment is frequently used to obtain superior mechanical properties by decomposing acicular α′ martensite into an equilibrium α + β phase. In this study, the microstructure and tensile properties of SLM Ti-6Al-4V alloy before and after various heat treatments were systematically investigated. The microstructure of the as-fabricated Ti-6Al-4V sample was composed of columnar prior β grains and acicular α′ martensite, which led to high strength (~1400 MPa) but low ductility (~5%) as well as significantly tensile anisotropy. The single heat treatment samples with lamellar α + β microstructure exhibited improved elongation to 6.8–13.1% with a sacrifice of strength of 100–200 MPa, while the tensile anisotropy was weakened. A trimodal microstructure was achieved through multi-step high-to-low-temperature (HLT) heat treatment, resulting in an excellent combination of strength (~1090 MPa) and ductility (~17%), while the tensile anisotropy was almost eliminated. The comprehensive mechanical properties of the HLT samples were superior to that of the conventional manufactured Ti-6Al-4V alloy.

## 1. Introduction

Titanium and its alloys have been extensively used in the aerospace, defense, marine, chemical industries, as well as in medical applications, due to their excellent properties, such as high specific strength, exceptional corrosion resistance, and good biocompatibility [1,2,3,4,5]. The toughness/ductility behavior of metallic materials is a main issue in many applications [6,7,8]. Ti-6Al-4V is an α + β alloy with high fracture toughness and outstanding comprehensive mechanical properties, which has become the most widely used titanium alloy to date [9,10,11]. However, the poor machinability of titanium alloys results in disadvantages such as low material efficiency, high production costs, and long production cycles when manufactured through traditional processing methods [1,4,12,13]. Selective laser melting (SLM), one of the most promising metal additive manufacturing (AM) processes, has received extensive attention since its emergence in the late 1980s and early 1990s [1,12,14]. SLM is guided by a computer-aided design (CAD) model to selectively melt metal powder layer by layer using a high-energy laser beam, enabling the rapid production of complex-shaped parts as a suitable replacement for traditional methods [15,16]. Furthermore, the unmelted powder can also be recycled, hence the material utilization rate is close to 100% in SLM.

Because of the layer-by-layer construction nature and extremely high cooling rate during the SLM process, columnar β grains growing along the building direction and extremely fine α′ martensite containing high-density defects will inevitably appear in the as-fabricated sample [17,18]. This results in high tensile strength (>1200 MPa) but low ductility (<10%), as well as significant mechanical property anisotropy between different directions [9,19]. Such microstructure characteristic limits the application of SLM Ti-6Al-4V [3]. Altering the morphology of β grains or transforming the brittle α′ martensite into a ductile α + β phase can effectively alleviate this issue. Kumar et al. [20] achieved short and discontinuous columnar grains in the as-fabricated samples by optimizing the SLM process parameters with layer thickness = 60 μm and scan rotation = 67°. This resulted in moderate ductility (~13%) and limited anisotropy, although the acicular martensite still existed in the sample. Xu et al. [21] achieved the in-situ decomposition of martensite into ultrafine lamellar α + β microstructures by adjusting the focus offset distance (FOD) within the range of 0 to 4 mm, which led to a high yield strength (1106 MPa) and moderate total elongation (11.4%). Fang et al. [4] studied the tensile properties of samples built in different directions and found that the as-fabricated samples exhibited strong tensile anisotropic behavior. After solution and aging treatment (SAT), the anisotropy was nearly eliminated, and the elongation of the samples exceeded 11% in all directions. Although the above studies had improved the comprehensive mechanical properties of SLM Ti-6Al-4V and significantly reduced the anisotropy by optimizing process parameters or SAT treatment, the plasticity of the samples was still lower than that of the conventional Ti-6Al-4V alloy [22,23,24].

Equiaxed microstructure is always desired because it can offer superior ductility and reduce the anisotropy of mechanical properties [9,25,26]. Hot working and subsequent annealing treatment are essential for globularizing α grain in conventional manufacturing [27,28]. However, introducing plastic deformation into the SLM forming process is challenging. Several recent studies have shown that an equiaxed microstructure can be achieved in SLM titanium alloys through appropriate heat treatment regimes. Zou et al. [29] obtained nearly equiaxed β-grains in SLM Ti-6Al-4V through rapid heat treatment, which increased the elongation from 5.2% to 16.6%. The formation of such a quasi-equiaxed structure was attributed to the epitaxial recrystallization during annealing. Zhang et al. [30] indicated that triple heat treatment could contribute to the globularization of lamellar α, resulting in the proportion of equiaxed α exceeding 25%. Due to the attainment of a trimodal microstructure, the samples exhibited outstanding comprehensive mechanical properties, with an ultimate tensile strength of 1019 MPa and an elongation of 16.3%, which was superior to that of the conventional Ti alloy. Sabban et al. [9] and Chen et al. [31] utilized an innovative cyclic heat treatment (CHT) method to optimize the microstructure. Since the repeated phase transformation process could provide an additional driving force for the spheroidization of α lath, a bimodal or nearly fully equiaxed microstructure was obtained. This led to a significant improvement in plasticity (>16%), with only a slight sacrifice in strength (>1000 MPa).

Until now, there have been no reports of achieving a trimodal microstructure in SLM Ti-6Al-4V alloy, which is considered to provide excellent comprehensive mechanical properties [32]. In addition, current research mainly focuses on the influence of single heat treatment on microstructure and properties, whereas the effect of heat treatment on the anisotropy of mechanical properties is rarely reported [16,17,33]. Therefore, this paper systematically investigates the effects of single and multi-step high-to-low-temperature (HLT) heat treatments on microstructure evolution, mechanical properties, and anisotropy behavior. The spheroidization mechanism of α lath and the role of HLT in eliminating the anisotropy of mechanical properties are also discussed.

## 2. Materials and Methods

### 2.1. Raw Materials

The raw material used in this study was gas-atomized Ti-6Al-4V powder obtained from Carpenter Technology Corporation (Philadelphia, PA, USA). Figure 1a–c shows the morphology and size distribution of the powder; it was mostly comprised of spherical particles with diameters ranging from 10 to 45 μm, and D10, D50, and D90 of the particles were measured as 21.81 μm, 33.4 μm, and 37.97 μm, respectively. Moreover, a small amount of satellite particles with a diameter less than 5 μm can occasionally be observed in larger particle powders (Figure 1b). The chemical composition (wt.%) of Ti-6Al-4V powder is listed in Table 1, which meets the standard specification of ASTM B348.

### 2.2. SLM Processing and Post-Heat Treatments

All Ti-6Al-4V samples were manufactured by an SLM machine (Concept Laser M-Lab) under the high-purity argon atmosphere to prevent oxidation. Based on the preliminary experiments, the optimized SLM processing parameters used in this study were as follows: the laser spot size was 50 μm, laser power (P) was 90 W, scanning speed (v) was 450 mm/s, hatch spacing (h) was 80 μm, and layer thickness (t) was set as 25 μm. The resulting volumetric energy density (E = P/vht) was determined to be 100 J/mm^3^. A zig-zag scan strategy was employed for printing each layer, i.e., the rotation angle between two successive layers is 0°, as shown in Figure 1d. Horizontal and vertical samples were fabricated to assess the impact of different building strategies on the anisotropy of tensile mechanical properties (Figure 1d). The cube samples had dimensions of 7 × 7 × 50 mm^3^, while the cylindrical samples were 7 mm in diameter and 50 mm in height. These samples will be hereinafter referred to as “AF-H” and “AF-V”, respectively.

To investigate the effects of various annealing schedules on microstructure and mechanical properties, part of the SLM Ti-6Al-4V samples were subjected to two types of heat treatments. One was the single heat treatment, i.e., samples were held at temperatures of 700 or 800 °C for 2 h followed by air cooling, which is hereinafter referred to as “HT700” and “HT800”, respectively. The other was multi-step high-to-low-temperature heat treatment, i.e., the sample was sequentially soaked at temperatures of 950, 850, 750, and 650 °C for 2 h, and each step from high to low temperature, as well as the subsequent cooling from 650 °C to room temperature, was carried out through furnace cooling (hereinafter referred to as “HLT”). The detailed information was illustrated in our previous work [34].

### 2.3. Microstructural Observation

Prior to microstructural observation, Ti-6Al-4V samples before and after heat treatments were cut by electric discharge machining (EDM). First, all samples were ground up to #2000 SiC sandpaper, then polished using diamond suspensions with particle sizes of 5 μm and 1.5 μm, and a solution consisting of 50 nm SiO_2_ suspension (OPS: 80 mL) and H_2_O_2_ (20 mL) was carried out for final mirror polishing. In this paper, X axis, Z axis, and Y axis are defined to represent the scanning direction (SD), building direction (BD), and transverse direction (TD) perpendicular to the BD and SD, respectively. XOZ plane was used for microstructural characterization, as shown in Figure 1d.

Backscatter electron (BSE) image was carried out on a scanning electron microscope (SEM, TESCAN MIRA 3 LMH) at an acceleration voltage of 20 kV. The electron backscatter diffraction (EBSD) test was captured using a TESCAN CLARA GMH scanning electron microscope equipped with an Aztec EBSD system operating at 15 kV under the working distance of 15 μm. The step sizes of 0.2 μm and 0.1 μm were used for low-magnification and high-magnification observation of AF samples, while the step sizes of 0.3 μm and 0.15 μm were used for low-magnification and high-magnification observation of post-heat treatment samples.

### 2.4. Mechanical Properties Characterization

Rod-like tensile samples were prepared parallel or normal to the BD (hereinafter referred to as “-V” and “-H”, respectively) with a gauge length of 15 mm and a diameter of 3 mm, which was based on GB/T 228.1-2021 standard [35] (Figure 1d). The room temperature tensile properties tests were performed on a universal testing machine (MTS, Eden Prairie, MN, USA, MTS E45.105) at a strain rate of 10^−3^ s^−1^, as measured by a contact extensometer. UTS, YS, El, and toughness were the average values of two repeated tests in each condition.

## 3. Results

### 3.1. Microstructures of as-Fabricated Sample

Figure 2 shows the microstructure of the as-fabricated Ti-6Al-4V sample. Acicular α′ martensite was the predominant morphology observed in the AF sample (Figure 2a), which exhibited a typical characteristic of SLM titanium alloys [2,13,16]. This can be ascribed to the extremely rapid cooling rate (~10^3^–10^8^ K/s) during the SLM process [13,36,37], significantly exceeding the critical cooling rate (410 K/s) required for martensite transformation in Ti-6Al-4V alloys [11,38]. In some acicular α′ martensite, high-density deformation twins with a thickness of 10–90 nm were mutually parallelly distributed (Figure 2b), which is mainly caused by residual stress due to the large temperature gradient within the molten pool and local strain arising from the β→α′ phase transformation [4,12].

The EBSD inverse pole figure (IPF) + image quality (IQ) map illustrates that the columnar prior β grains were grown epitaxially along the building direction, and the average width of the columnar prior β grain was 88 μm (Figure 2c). During the layer-by-layer deposition process, the laser not only melts the current powder layer, but also penetrates through several previously solidified layers, causing them to remelt and form a molten pool [2]. Moreover, there is a steep thermal gradient between the center and edge of the molten pool, and the direction of heat dissipation within a molten pool is opposite to the building direction, ultimately leading to the preferential growth of the prior β grain [10,39]. Acicular α′ martensite with hierarchical structure was embedded in the columnar prior β grain due to the complex thermal history during the forming process (Figure 2c), as widely reported in other literature [3,4,12]. According to their sizes, these α′ martensites can be classified as primary, secondary, tertiary, and quartic α′ martensites. It can be seen in Figure 2c that the primary α′ martensites with the same orientation was parallel to each other and extended throughout the entire prior β grain. The length and width of primary α′ martensites were in the range of 20–120 μm and 1–2 μm, respectively. Fine secondary α′ martensite was distributed parallel or perpendicular to the primary α′ martensite (Figure 2d). Its length and width were significantly reduced, measuring only 5–15 μm and 0.4–1 μm, respectively. The appearance of primary α′ martensite is earlier than that of secondary α′ martensite [3], thus constraining the growth of the latter. Furthermore, even finer tertiary and quartic α′ martensites can be distinguished in the EBSD image. Figure 2d confirms the existence of nano-twins and further reveals that nano-twins were predominantly present within the primary α′ martensite, with a minor distribution in part of the secondary α′ martensite. The proportions of high-angle grain boundaries (HAGBs) and low-angle grain boundaries (LAGBs) in the AF sample were 88.6% and 11.4%, respectively (Figure 2e). Kernel average misorientation (KAM) values can be utilized to estimate the geometrically necessary dislocations (GNDs) density [31,40]. Figure 2f shows that the distribution of KAM was inhomogeneous, and the KAM values in tertiary and quartic α′ martensites were notably higher than those in the primary and secondary α′ martensites, implying that finer α′ martensite contains a higher density of dislocations [15].

### 3.2. Microstructures of Single Heat Treatment Samples

Figure 3 shows the microstructures of the as-fabricated Ti-6Al-4V samples after single heat treatment at different temperatures. As can be seen from Figure 3a,d, the original columnar boundaries still existed in the HT700 and HT800 samples. After annealing at 700 °C for 2 h, the microstructure was predominantly composed of a fine needle-like α phase (Figure 3a,b). Moreover, these α grains also exhibited a hierarchical structure, which originated from different sizes of martensite present in the AF sample. The widths of the primary and secondary α grains were 1.2–2.3 μm and 0.6–1.2 μm, respectively, showing a slight coarsening compared to the AF sample. Martensite is a metastable phase containing a large amount of supersaturated β stabilizing elements (V). When the heat treatment temperature is higher than the martensite decomposition temperature (~400 °C) [21,36], the α phase begins to nucleate at the α′ martensite grain boundary and grows along the martensite. In the meanwhile, the β stabilizing element is continuously expelled from the α′/α′ boundaries, forming a V-enriched region, where the β phase will subsequently nucleate and grow [2,12,30]. The aforementioned process results in the decomposition of martensite into equilibrium α + β phases. The presence of fine β particles or films formed along the needle α boundaries further confirmed the decomposition of α′ martensite, as indicated by the arrows in Figure 3b,c. Simultaneously, a small amount of β particles was observed within the α phase (Figure 3c). The formation of β particles is due to the AF sample prepared by this experiment, containing numerous lattice defects such as twins and dislocations (Figure 2). During the annealing process, the β phase tends to precipitate at these lattice defects [41]. It is worth noting that nano-twins still existed within some α grains (Figure 3c).

When the heat treatment temperature increases to 800 °C, a typical lamellar α + β microstructure was obtained as a result of the initial acicular martensite decomposition and subsequent coarsening (Figure 3d–f). Similarly, the hierarchical lath α inherited from α′ martensite was also observed, in which the widths of primary and secondary α-laths were 1.3–2.4 μm and 0.8–1.5 μm, respectively. Compared to the HT700 sample, the α thickness of HT800 had a certain coarsening, owing to the fact that the interfacial energy decreased and the atomic diffusion rate enhanced with increasing annealing temperature [17,39]. The β phase with an average thickness of about 80 nm tended to connect into films and grew up along the α-lath boundaries, as indicated by the arrows in Figure 3e,f. The nano-twins that previously existed in the AF sample were not detected after annealing at 800 °C for 2 h. This is consistent with the previous literature [33]. Note that the semi-continuous grain boundary α phase (GB-α) formed along the prior β grain (Figure 3d), which may have a negative effect on the tensile elongation of the sample [19]. Additionally, some of the α laths were fractured due to the segmentation of β film, resulting in the formation of a few equiaxed α grains, as shown in Figure 3f.

Figure 4 shows the EBSD analysis results of single heat treatment samples. The IPF + IQ map confirms the existence of prior β grains in HT700 and HT800 samples (Figure 4a,c), and their widths had not changed significantly, measuring 85 μm and 89 μm, respectively. When the heat treatment temperature is below the β transus temperature, the undissolved α phase plays a role in pinning the β grain boundaries, so that its structure can be retained [12,30]. Thus, the columnar structure can only be eliminated when the temperature exceeds the β transus temperature [32,42]. In the HT800 sample, the equiaxed α grains were observed to have preferentially formed along the prior β boundaries (Figure 4c). Furthermore, the annealing treatments resulted in a reduction of LAGBs in the HT700 and HT800 samples to 6.37% and 6.63% (Figure 4b,d), respectively.

### 3.3. Microstructures of Multi-Step High-to-Low-Temperature Heat Treatment Sample

The above single heat treatments resulted in a full α + β lamellar microstructure, with only a small amount of α laths spheroidized and transformed into equiaxed grains. However, the proportion of equiaxed grains was too small to produce a significant impact on improving the overall mechanical properties. The latest research suggests that multiple annealing at different temperatures can effectively spheroidize the microstructure [25,30,32]. Therefore, a HLT treatment regime was proposed to further optimize the microstructure.

Figure 5 shows the microstructure of the as-fabricated Ti-6Al-4V sample after HLT treatment. Since the maximum temperature of HLT was below the β transus temperature, the original columnar grain with an average width of 90 μm could still be observed (Figure 5a). The high-magnification BSE image in Figure 5b illustrates that the microstructure after HLT treatment was mainly composed of lamellar, short-rod, and equiaxed α grains, showing a significant change compared with that of HT700 and HT800 samples. Thus, a trimodal microstructure was achieved through HLT treatment. The average width of the lamellar α was 2 μm, while the short-rod and equiaxed α exhibited a thickness in the range of 1.6–2.5 μm and 3–6 μm, with an aspect ratio of ~0.37 and ~0.53, respectively. Moreover, β layers with an average thickness of 0.27 μm formed between the α grains.

Further microstructural analysis of the HLT sample was conducted using EBSD. The IPF + IQ map in Figure 5c reveals that equiaxed α grains originated from three different locations and exhibited distinct spheroidization behaviors. The first type of equiaxed grain originated from the prior β grain boundaries. Typically, subcritical annealing often results in the formation of continuous grain boundary α phase (GB-α), as widely reported in previous studies [25,43,44,45]. This is due to the activation energy for the α phase nucleated at the prior β grain boundaries being lower than at other sites, which leads to the numerous α grains preferentially forming and growing to connect with each other [43,44], whereas in the present study, the α phase at the prior β grain boundaries mainly had an equiaxed morphology, and the continuous GB-α was absent. On the one hand, when the HLT sample cooled from 950 °C to 850 °C, some lamellar α phases with different orientations would epitaxially grow into the prior β grain boundaries, thereby separating the GB-α phases from each other [45], as indicated by the white arrows in Figure 5c,d. On the other hand, there was a high dislocation density at the prior β grain boundaries vicinity (Figure 2f), which could provide a strong driving force for recrystallization [46]. As a result, the newly formed GB-α readily underwent spheroidization and transformed into equiaxed grains. The second type of equiaxed grain was derived from the fracture of the primary and secondary α phase by boundary splitting. The microstructure of the AF Ti-6Al-4V sample consisted of acicular α′ martensite, which contained high-density dislocations (Figure 2f). During the high-temperature annealing process, these dislocations would be fully activated and gradually evolved into planar dislocation arrays, thus reducing the total Gibbs energy of the system [47,48]. This resulted in the formation of an initial dislocation substructure [25]. With the extension of annealing time, some dislocation arrays could further develop into stable sub-grain boundaries within the lamellar α through polygonization [25,30,31]. The β phase formed by martensite decomposition mainly existed in a layered morphology at the boundaries of the α phase. These β phases intersected with adjacent α lamellae and the sub-grain boundaries within the lamellae, forming a so-called triple junction [9,49]. Due to the instability of the triple junction, thermal grooving was prone to form in this area. Once the grooving was generated, the β phase would continuously penetrate into the grooving until it completely broke up the lamellae [27,28], as indicated by the black arrows in Figure 5d. This process is referred to as thermal grooving and boundary splitting, which is a common spheroidization mechanism in both conventional and additive manufacturing titanium alloys. In addition to sub-grain boundaries, the nano-twins inherently presented in α′ martensite (Figure 2b,d) could also lead to the occurrence of thermal grooving and boundary splitting [34]. It is worth noting that the proportion of LAGBs in the HLT sample was significantly higher than that in the HT700 and HT800 samples (Figure 5e), with a proportion up to 11%. This could be attributed to plastic strain resulting from lattice mismatch and volume change during the β→α phase transition process [31]. The KAM value at the LAGBs was notably higher compared to other positions (Figure 5f), which provided evidence for the existence of local plastic deformation. These LAGBs could facilitate the penetration of the β phase into the grooving and promote early spheroidization [50]. The third type of equiaxed grain originated from the direct spheroidization of the tertiary and quartic α′/α phase by cylinderization. The curvature at the edge of the original microstructure was relatively high, and these regions contain high-density dislocations (Figure 2). As a result, the tip would transform into a flat surface to reduce the surface energy, and this process is known as cylinderization [9]. Once the boundary splitting and cylinderization were completed, termination migration and Oswald ripening would occur and lead to the final spheroidization [27,28,51].

### 3.4. Tensile Properties

Figure 6a–c shows the tensile properties of as-fabricated and various heat treatments for SLM Ti-6Al-4V samples. In order to investigate the influence of different orientations on tensile properties, samples in both horizontal and vertical directions were tested. The average values of UTS, YS, El, and toughness are summarized in Table 2. As can be seen from the tensile engineering stress–strain curves in Figure 6a, the AF samples exhibited limited work-hardening capability, which resulted in a low uniform elongation, and the value of AF-H was only 4.0 ± 0.55%. After annealing at 700 °C for 2 h, the work-hardening capability showed marginal change. When the temperature was increased to 800 °C, the work-hardening capability significantly improved, and the uniform elongation of HT800-H rose to 6.1 ± 0.04%. For the sample after HLT treatment, the work hardening ability and uniform elongation (8.3 ± 0.46%) reached the maximum value. Furthermore, the work-hardening capability of all samples in the horizontal direction was significantly higher than in the vertical direction. For instance, the uniform elongation in the vertical direction for as-deposited and HT800 samples was only 1.7 ± 0.08% and 1.8 ± 0.20%, respectively.

AF samples showed the highest UTS (1368–1418 MPa) and YS (1265–1400 MPa), but a relatively low El of about ~5%. The brittleness can be attributed to the very fine acicular martensite and its internal high-density dislocations and twins. In addition, there was a significant anisotropy of mechanical properties between the horizontal and vertical directions. To quantitatively calculate the degree of anisotropy in the samples, the following formula was employed [26]:(1)$$Ix=\left|xH-xV\right|/\bar{x}(0\leq Ix<2)$$
where x represented UTS, YS, or El, and $\bar{x}$ denoted the mean value of the properties in the two directions. The calculation results are listed in Table 3. With the increase of the Ix value, the degree of anisotropy becomes greater. Apparently, the anisotropy of elongation was more pronounced in the AF samples (I_El_ = 0.766). The single annealing treatments resulted in a decreased strength of 100–200 MPa, but the elongation increased from 6.8–8.5% for HT700 samples and 8.9–13.1% for HT800 samples. The annealing treatments significantly reduced the anisotropy level in the elongation of HT700 and HT800. Although the HLT treatment led to a decrease in the UTS to 1089–1092 MPa, the presence of considerable equiaxed grains in the HLT samples resulted in a significant increase in elongation, which exceeded 15%. Especially in the case of HLT-V, the elongation increased by nearly two times compared to HT800-V. Note that the anisotropy of UTS, YS, and El had been almost eliminated after HLT treatment (I_UTS_ = 0.003, I_YS_ = 0.047, I_El_ = 0.157). Toughness can be used to estimate the comprehensive mechanical properties of materials. In the present work, the toughness values of the samples before and after heat treatments were obtained by the integration of the engineering stress–strain curves (Figure 6c). It can be seen that the AF samples had the lowest toughness value. With the increase in heating temperature, the toughness value gradually improved. After the HLT treatment, the toughness value reached the maximum in both directions (> 160 MJ/m^3^), which meant that the HLT samples had the optimal comprehensive mechanical performance.

Based on tensile properties data available in the literature, a comparison of the UTS vs. El between conventional manufacturing [22,23,24,52,53] and SLM [4,9,29,54,55] Ti-6Al-4V components as well as the present study is shown in Figure 6d. The conventional Ti-6Al-4V samples exhibited moderate strength (850–1000 MPa) and high elongation (12–18%), while SLM Ti-6Al-4V exhibited higher strength (1150–1350 MPa) and lower elongation (3–10%). Although the post-heat treatment had a certain sacrifice in strength, it could significantly improve ductility and reach a level comparable to conventional Ti-6Al-4V. In this study, the AF samples presented higher strength but slightly lower elongation compared to other SLM Ti-6Al-4V components. Although single heat treatments improved the plasticity of the material, it was still lower than the conventional level. Owing to the attainment of a trimodal microstructure, the HLT samples exhibited the best combination of strength and ductility, and their mechanical performance was superior to conventionally manufactured Ti-6Al-4V.

## 4. Discussion

In this study, a trimodal microstructure composed of lamellar, short-rod, and equiaxed α grains was achieved through HLT treatment, resulting in both high strength and high ductility, while the anisotropy of mechanical properties was almost eliminated. The effects of HLT treatment on microstructure evolution and the influence of microstructure on mechanical properties and anisotropic behavior are discussed as follows.

### 4.1. The Function of Temperature in Each Step of HLT Treatment

In conventional Ti-6Al-4V alloy, martensitic or fully lamellar α + β microstructures typically result in poor ductility. Therefore, transforming the partial or entire lamellar structure into equiaxed grains can significantly improve the comprehensive performance of the material and meet the requirements of engineering applications [24,53]. The globularization of conventional titanium alloys usually involves hot working and subsequent annealing treatment [28,56,57]. Apparently, significant plastic deformation is a prerequisite for globularization of the microstructure because the recrystallization process requires a driving force. Because globularization is a thermally activated and diffusion-controlled process, spheroidization behavior exhibits a high temperature dependence; that is, only a sufficient temperature can enable complete recrystallization to occur [27].

However, due to the fact that SLM is a near-net-shape manufacturing technology, it is not feasible to undergo a plastic deformation process. Thanks to the extremely high cooling rate during formation, SLM parts usually contain high-density dislocations and twins. These substructures are similar to those in the deformed sample, which provides the possibility for the spheroidization of the microstructure. Currently, several studies are focusing on modifying the microstructure of additive manufacturing components through post-heat treatment. Liu et al. [1] and Cao et al. [33] demonstrated that annealing at subcritical temperature could result in equiaxed grains in the sample, owing to the boundary splitting, whereas only a small amount of α grains underwent spheroidization. Therefore, more advanced heat treatment is required to further globularize the microstructure. Zhao et al. [25] achieved a bimodal microstructure in LSF Ti-6Al-4V-ELI alloy through triple heat treatment with equiaxed grains accounting for 78%, which resulted in an improved ductility of about 25%. Chen et al. [31] and Sabban et al. [9] obtained over 50% equiaxed grains in SLM Ti-6Al-4V-ELI by cyclic heat treatment (CHT). In fact, the fraction of equiaxed grains reached 100% in the intermediate state. To our knowledge, achieving such a high percentage of equiaxed grains in SLM Ti-6Al-4V alloy has not been reported. This can be attributed to the fact that Ti-6Al-4V-ELI alloy contains fewer interstitial solid solution atoms (C, N, O), and dislocations are more easily activated and form subgrain boundaries during subcritical annealing [25]. In this study, a trimodal microstructure was obtained in SLM Ti-6Al-4V alloy for the first time by using the proposed HLT heat treatment regime, and the comprehensive mechanical properties of the material were significantly improved. Thus, it is necessary to clarify the function of each temperature stage of HLT treatment on the microstructure evolution.

Based on the aforementioned information, the primary temperature is crucial for spheroidization behavior. Therefore, we chose 950 °C as the starting temperature to ensure that dislocations could be fully activated. When held at this temperature, α′ martensite decomposed and transformed into the equilibrium α + β phase, while the activated dislocations gradually evolved into subgrain boundaries within the α lamellae. A triple junction formed at the intersection between the subgrain boundary and α/β interface, consequently weakening the α boundary. As annealing time extended, the α lamellae eventually fractured into several fragments through the penetration of the β phase, resulting in the initial globularization. In this case, intragranular twins could also play a similar role as subgrain boundaries to promote the splitting of α lamellae. According to the chemical composition of the Ti-6Al-4V sample in this experiment, the variation curve of the phase fraction with temperature was obtained via JmatPro 7.0 software, as shown in Figure 7. It can be seen from the phase diagram that β was the dominant phase at 950 °C, thus the α→β transition also occurred simultaneously. In this process, the β phase nucleated at intragranular defects (Figure 3) may potentially facilitate the fracture of α lamellae through epitaxial growth. Furthermore, the α phase globularized through cylinderization and α nucleated at the prior β boundary were also in progress. Upon cooling from 950 °C to 850 °C, the content of the β phase gradually decreased from 72.7% to 30.1% (Figure 7). Part of the β phase would be added to the pre-existing equiaxed α, leading to its epitaxial growth, while the other part of the β phase would transform into lamellar α. During this process, approximately 40% of the β phase transformed into the α phase, resulting in significant lattice mismatch and volume changes. Therefore, a large number of dislocations would accumulate within the α lamellae. By prolonging the annealing time at 850 °C, these dislocations would evolve into subgrain boundaries, leading to the fracture of α lamellae and forming the short-rod α. In addition, as mentioned in Section 3.3, these epitaxially grown lamellar α phases also played a role in separating the GB-α, thereby promoting the formation of equiaxed grains at the prior β boundaries. As the annealing temperature further decreases to 750 °C and 650 °C, the grain size of equiaxed, short-rod, and lamellar α continuously increased with the transformation of β→α. Ultimately, a trimodal microstructure was formed by these processes.

### 4.2. Influence of Microstructure on Mechanical Property and Anisotropic Behavior

Due to the extremely high cooling rate during the SLM process, fine acicular α′ martensite containing high-density dislocations and nano-twins will inevitably appear in the AF Ti-6Al-4V sample, which results in high strength but relatively poor ductility. Moreover, the building process is always accompanied by a large temperature gradient, and columnar grains are prone to form along the deposition direction, which is considered to be a major factor in the anisotropy of mechanical properties [10,11,17]. Owing to the large aspect ratio of columnar grains, AF-H contained more columnar grains per unit length than AF-V along the tensile direction [4]. When subjected to the same strain, individual columnar grains in AF-V undertook more plastic deformation, consequently resulting in more significant stress concentration. Once plastic deformation could not be coordinated, the voids would nucleate in the stress concentration region [58]. As the stretching proceeds, the voids gradually coalesced and led to the final fracture. This explains why the plasticity of the AF-H is superior to that of AF-V. In addition, the cooling rate along the building direction is higher than that of the scanning direction, so the sample in the vertical direction has a finer microstructure [39]. This may be another reason for the higher strength of AF-V. Due to the elimination of dislocations and twins, the strength of the HT700 and HT800 samples decreased by 100–200 MPa. Compared with the AF sample, the width of the α lath was slightly coarsened. As a result, the effective slip length within the laths increases, which is more conducive to plastic deformation [59]. The decomposition of α′ martensite led to the formation of the β phase along the boundary of α laths. The β films between the α grains can act as a lubricant to effectively transmit stress and accommodate plastic deformation [58,59]. Thus, the plasticity of HT700 and HT800 samples was improved. Although the annealing treatments failed to eliminate the columnar structure, the anisotropy in the horizontal and vertical directions was weakened due to the improved compatibility of deformation and the effective release of stress concentration during the deformation process. HLT treatment further coarsened the α grains, which led to an increase in effective slip length compared to single annealing treatments. Furthermore, the content of the β phase in HLT (15.93%) was almost twice as much as that of HT800 (8.33%), hence the lubricating effect of the β phase would be more pronounced. More importantly, HLT treatment resulted in the formation of equiaxed α grains both at the prior β grain boundaries and intragranular. Owing to the lamellae, short-rod, and equiaxed grains in the trimodal structure contributing to strengthening and toughening, the HLT samples achieved excellent comprehensive mechanical properties. In a recent report, the discontinuous GB-α could fully accommodate plastic deformation in both loading directions and effectively reduce the mechanical property anisotropy [19]. Combined with the isotropic structural characteristics of equiaxed grains in this work, the HLT sample exhibited almost isotropic behavior.

## 5. Conclusions

In this paper, the microstructure and tensile properties of the SLM Ti-6Al-4V alloy before and after various heat treatments were systematically studied. The formation mechanism of equiaxed grains in HLT treatment was analyzed in detail, and the relationship between microstructure and mechanical properties as well as anisotropic behavior was also discussed. The main conclusions were as follows:(1)The microstructure of the as-fabricated sample was characterized by columnar prior β grains and acicular α′ martensite with a hierarchical structure. Due to the uneven microstructural features, high-density dislocations, and nano-twins, the AF sample exhibited high strength (1368–1418 MPa) but low ductility (3.3–7.4%), as well as significant mechanical anisotropy. For example, the elongation in the horizontal direction was 124% higher than that in the vertical direction.(2)Single heat treatments decomposed α′ martensite into equilibrium α + β phase, and the width of α laths slightly coarsened with increasing temperature. The elimination of high-density defects resulted in improved ductility (6.8–13.1%), accompanied by a strength sacrifice of 100–200 MPa. Furthermore, the anisotropy in mechanical properties significantly decreased.(3)A trimodal microstructure was achieved through HLT treatment. Owing to the lamellae, short-rod, and equiaxed grains contributing to strengthening and toughening, the HLT sample exhibited optimal comprehensive mechanical performance with UTS of 1089–1092 MPa and El of 15.8–18.5%, which was superior to that of the conventional Ti-6Al-4V alloy. The formation of equiaxed grains in both prior β boundaries and intragranular, as well as the lubricating effect of β films, significantly improved the accommodated deformation ability, thus obtaining an almost isotropic sample after HLT treatment.(4)HLT treatment could contribute to fully activating dislocations and provide additional driving force for spheroidization behavior. The globularization mechanism of α grains was attributed to boundary splitting and cylinderization.

## Figures and Tables

**Figure 1 materials-16-06947-f001:**
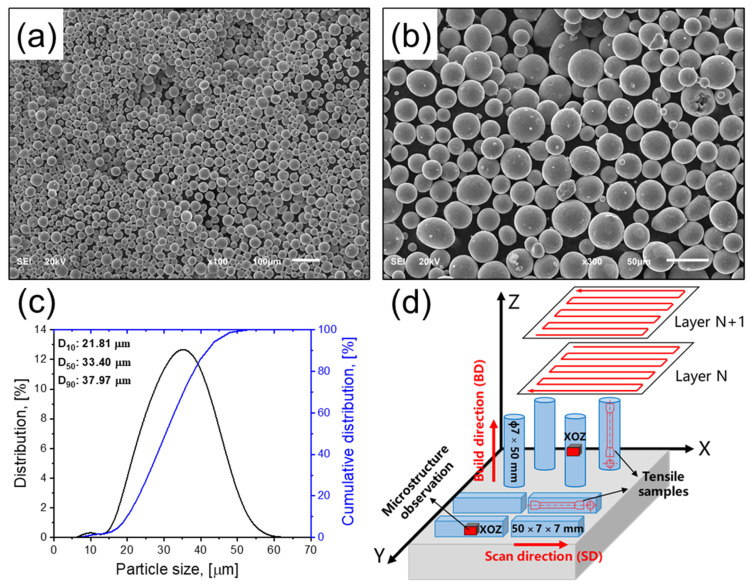
Ti-6Al-4V powder (**a**,**b**) morphology with different magnifications and (**c**) particle size distribution. (**d**) schematic diagram of SLM processing and manufactured samples.

**Figure 2 materials-16-06947-f002:**
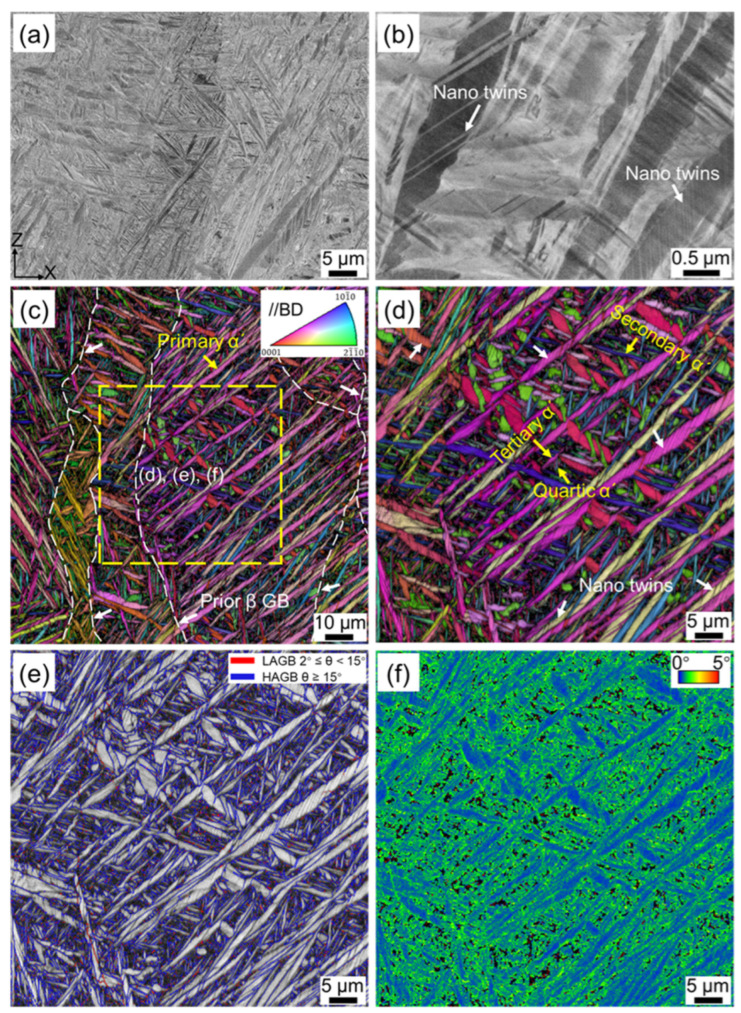
Microstructures of as-fabricated SLM Ti-6Al-4V sample: (**a**) BSE image, (**b**) high-magnification BSE image, (**c**) EBSD inverse pole figure (IPF) + image quality (IQ) map, (**d**) high-magnification IPF + IQ map, and (**e**) grain boundary (GB), and (**f**) kernel average misorientation (KAM) of yellow dashed box in (**c**).

**Figure 3 materials-16-06947-f003:**
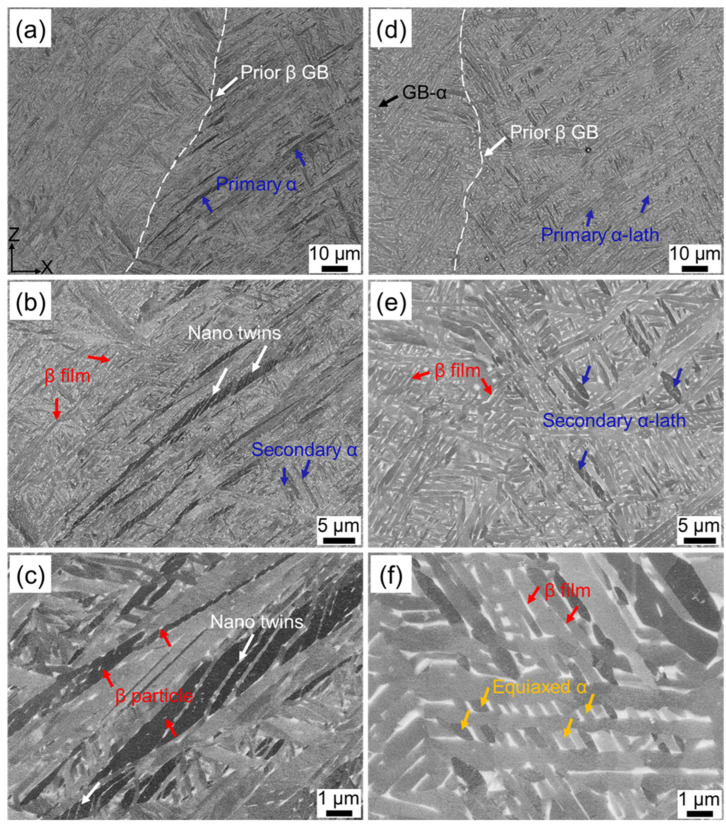
BSE images with different magnifications showing the microstructure of SLM Ti-6Al-4V after various single heat treatments: (**a**–**c**) HT700, (**d**–**f**) HT800.

**Figure 4 materials-16-06947-f004:**
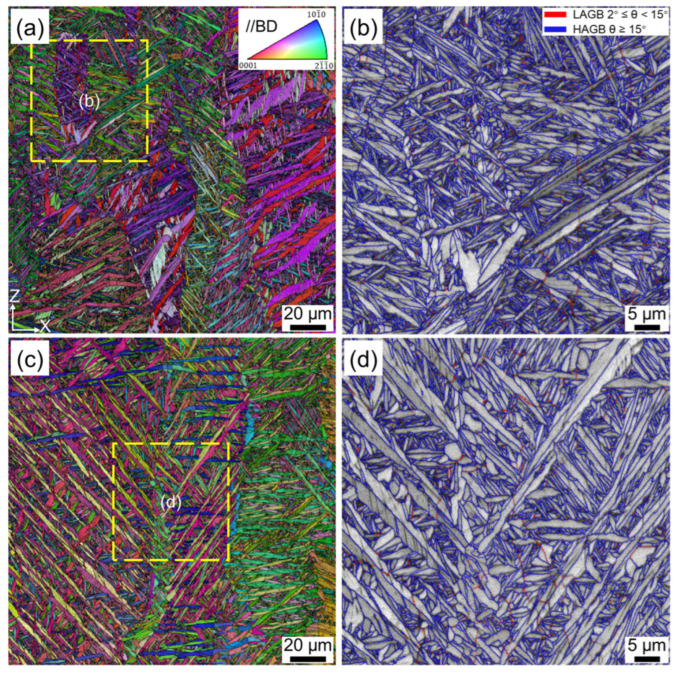
EBSD IPF + IQ maps and GB maps of SLM Ti-6Al-4V after various single heat treatments: (**a**,**b**) HT700, (**c**,**d**) HT800.

**Figure 5 materials-16-06947-f005:**
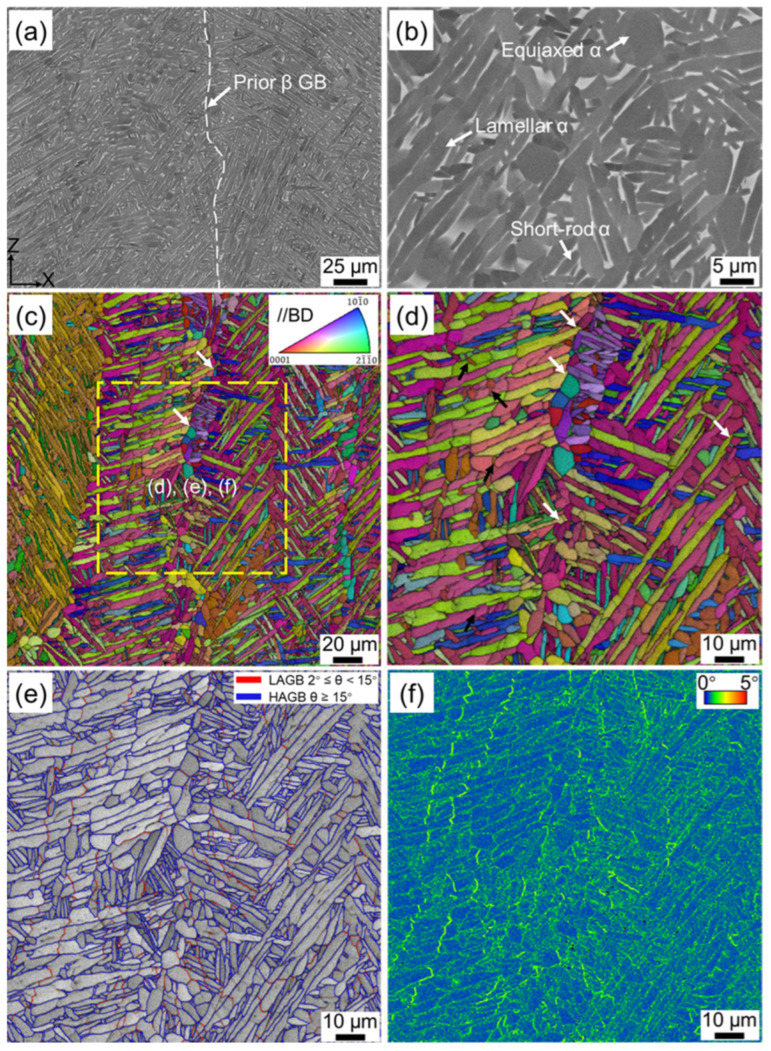
Microstructures of SLM Ti-6Al-4V after HLT treatment: (**a**) BSE image, (**b**) high-magnification BSE image, (**c**) EBSD IPF + IQ map, (**d**) high-magnification IPF + IQ map, (**e**) GB, and (**f**) KAM of yellow dashed box in (**c**).

**Figure 6 materials-16-06947-f006:**
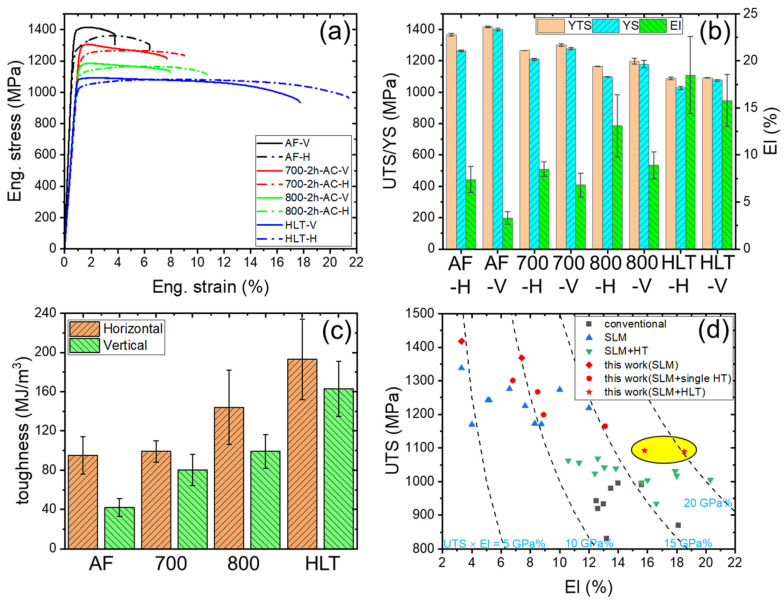
(**a**) Engineering tensile stress–strain curves, (**b**,**c**) tensile properties, (**d**) a comparison of the UTS vs. El between conventional manufacturing and SLM Ti-6Al-4V alloy.

**Figure 7 materials-16-06947-f007:**
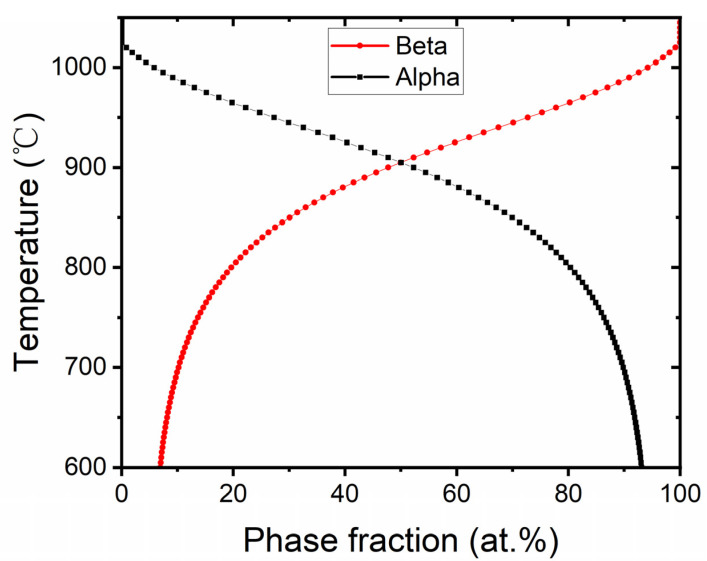
The variation curve of the phase fraction with temperature was obtained via JmatPro 7.0 software for the Ti-6Al-4V alloy.

**Table 1 materials-16-06947-t001:** Chemical composition of the Ti-6Al-4V powder (wt.%).

Element	Ti	Al	V	Fe	O	C	N	H
ASTM B348	Balance	5.5–6.75	3.5–4.5	0.40	0.20	0.08	0.05	0.015
Measured values	Balance	6.52	4.27	0.20	0.134	0.05	0.04	0.0027

**Table 2 materials-16-06947-t002:** Measured mechanical properties data of SLM Ti-6Al-4V under different conditions.

Samples	UTS (MPa)	YS (MPa)	El (%)	Toughness (MJ/m^3^)
AF-H	1368 ± 9	1265 ± 5	7.4 ± 1.4	95 ± 19
AF-V	1418 ± 4	1400 ± 8	3.3 ± 0.7	42 ± 9
HT700-H	1267 ± 0	1210 ± 6	8.5 ± 0.8	99 ± 11
HT700-V	1301 ± 8	1279 ± 7	6.8 ± 1.3	80 ± 16
HT800-H	1165 ± 2	1100 ± 1	13.1 ± 3.3	144 ± 38
HT800-V	1199 ± 18	1180 ± 24	8.9 ± 1.4	99 ± 17
HLT-H	1089 ± 8	1028 ± 9	18.5 ± 4.1	193 ± 41
HLT-V	1092 ± 2	1077 ± 5	15.8 ± 2.8	163 ± 28

**Table 3 materials-16-06947-t003:** Tensile anisotropic index (I) of SLM Ti-6Al-4V under different conditions.

Samples	I_UTS_	I_YS_	I_El_
AF	0.036	0.101	0.766
HT700	0.026	0.055	0.222
HT800	0.029	0.070	0.382
HLT	0.003	0.047	0.157

## Data Availability

The raw data required to reproduce these findings are available from the corresponding author upon reasonable request.

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
