# Peer review of "Achieving an Excellent Strength and Ductility Balance in Additive Manufactured Ti-6Al-4V Alloy through Multi-Step High-to-Low-Temperature Heat Treatment"

_materials, 2023, doi:10.3390/ma16216947_

Round 1
Reviewer 1 Report
Comments and Suggestions for Authors
The study presented in this paper is on the use of selective laser melting (SLM) as an effective alternative to traditional machining methods to prepare parts of arbitrary and complex shapes by gradually accumulating layers of Ti-6Al-4V alloy. The microstructure and mechanical properties of Ti-6Al-4V alloy produced by SLM before and after various heat treatments were analyzed. The authors of the publication conclude that advanced heat treatments can significantly improve the mechanical properties of SLM-produced alloys, and after multistage variable-temperature heat treatment (HLT), a trimodal microstructure was obtained, resulting in an excellent combination of strength (~1090 MPa) and ductility (~17%), while almost completely eliminating tensile anisotropy. The overall mechanical properties of the HLT samples were better than those for conventionally fabricated Ti-6Al-4V alloy, opening up new prospects for industrial applications of these materials. In terms of microstructural analysis, the topics of the problem analyzed in great detail in depth.
On the other hand, some questions in the field of testing of mechanical properties are raised by the lack of reference to the effect of scale of the test specimens as well as justification for the selection of the dimensions of samples for tensile testing, which clearly deviate from the recommendations of norms and standards in this regard. Clarification of this point will further enhance the substantive value of this article
Comments on the Quality of English Language Please have a technical native speaker check
Reviewer 2 Report
Comments and Suggestions for Authors
The paper deals with strength/ductility improvement in SLM Ti6Al4V. This is an important topic and the paper is in general well written. I suggest some minor revisions before accepting the paper:
1. References are mainly by Chinese people. I suggest to update with some references also by other countries researchers.
2. Line 31: before writing about Ti6Al4V fracture toughness behavior I suggest the following more general sentence:
"The toughness/ductility behavior of metallic materials is a main issue in many application [5,6]".
[5] Kumar, M.; Neelakantha, V. IOP Conf. Serr: Materials Science and Engineering, 2021, 1013 012010
[6]Di Schino, A.; Guarnaschelli, C. Materials Science Forum, 2010, 638-642, 3188-3193. https://doi.org/10.4028/www.scientific.net/MSF.638-642.3188
Line 87-91: I suggest to write at present time (not past):
Therefore, this paper systematically investigates the effects of single and multi-step high-to-low-temperature (HLT) heat treatments on microstructure evolution, mechanical properties, as well as anisotropy behavior. The spheroidization mechanism of α lath and the role of HLT in eliminating the anisotropy of mechanical properties are also discussed.
Reviewer 3 Report
Comments and Suggestions for Authors
The reviewer would like to appreciate the authors for presenting the work very well. The results are well presented in a proper pattern.
The authors are requested to answer the following queries to improve the quality of the paper.
1. Why was only SLM used to fabricate the parts? Can we expect similar results, when the same component is manufactured using any other AM technique?
2. The authors are suggested to present the pictures of the AM fabricated components.
3. In section 2.2, SLM parameters are presented. The authors are required to correlate the processing parameters with the macroscopic images of the AM parts fabricated.
4. The authors are suggested to give a detailed explanation of the reason behind the selection of specific heat treatment parameters.
5. The authors are required to check the manuscript for grammatical mistakes.
Comments on the Quality of English Language
The authors are required to check the manuscript for grammatical mistakes.
